# Glutamate production from aerial nitrogen using the nitrogen-fixing bacterium *Klebsiella oxytoca*
Daisuke Yoshidome [1] ✉, Makoto Hidaka[1], Toka Miyanaga[1], Yusuke Ito[1,2], Saori Kosono[1,3,4] & Makoto Nishiyama [1,3] ✉

Glutamate is an essential biological compound produced for various therapeutic and nutritional applications. The current glutamate production process requires a large amount of ammonium, which is generated through the energy-consuming and $CO_2$-emitting Haber–Bosch process; therefore, the development of bio-economical glutamate production processes is required. We herein developed a strategy for glutamate production from aerial nitrogen using the nitrogen-fixing bacterium *Klebsiella oxytoca*. We showed that a simultaneous supply of glucose and citrate as carbon sources enhanced the nitrogenase activity of *K. oxytoca*. In the presence of glucose and citrate, *K. oxytoca* strain that was genetically engineered to increase the supply of 2-oxoglutarate, a precursor of glutamate synthesis, produced glutamate extracellularly more than $1 \, g \, L^{-1}$ from aerial nitrogen. This strategy offers a sustainable and eco-friendly manufacturing process to produce various nitrogen-containing compounds using aerial nitrogen.

Glutamate is a direct precursor of various nitrogen-containing compounds, such as arginine, ornithine, proline, and glutathione, and most amino acids in living organisms are synthesized through the direct or indirect transfer of the amino group of glutamate[1]. Therefore, glutamate is one of the most biologically essential compounds in all living organisms. It is also used in medical fields as a clinical nutrient in amino acid infusions and oral/enteral nutritional supplements and as a component of pharmaceuticals, such as therapeutic agents for psychiatric disorders and fatigue[2,3]. In addition, glutamate is an umami compound that enhances the taste of food and improves nutrition[4]. To meet the demand for practical applications, a large amount of glutamate (3.3 million tons (Tg) $yr^{-1}$) is produced industrially using microorganisms[5]. Glutamate fermentation was initially established by Kinoshita and Udaka using the aerobically grown bacterium *Corynebacterium glutamicum*[6]. However, this process requires a large amount of ammonium and glucose as nitrogen and carbon sources, respectively, which are directly or indirectly dependent on chemically fixed nitrogen of 120 Tg $yr^{-1}$ via the Haber–Bosch process[7], which is highly energy-demanding (~2% of total energy) and emits large amounts of $CO_2$[8]. To produce 3.3 Tg $yr^{-1}$ glutamate, ~0.31 Tg $yr^{-1}$ of fixed nitrogen and 3.84 Tg $yr^{-1}$ of sugar are stoichiometrically required as raw materials. Fixed nitrogen of 4.03 Tg $yr^{-1}$ is applied as nitrogen fertilizer to sugar crops worldwide[9] in

order to produce around 170 Tg $yr^{-1}$ of sugar in the world[10]. Therefore, 0.095 Tg $yr^{-1}$ fixed nitrogen, which corresponds to 0.08% of total Haber–Bosch demands, is estimated to be indirectly consumed for glutamate fermentation via its carbon source, while 0.31 Tg $yr^{-1}$ fixed nitrogen as a nitrogen source for glutamate fermentation corresponds to 0.26%. In total, 0.34% of nitrogen chemically fixed by the Harber-Bosch process is used for current glutamate fermentation. Although this value appears to be low; it only reflects glutamate fermentation; therefore, a sustainable and eco-friendly bioprocess independent of the Harber-Bosch process needs to be established in the near future.

Nitrogen-fixing bacteria (diazotrophs) specifically produce molybdenum-containing nitrogenase, encoded by *nifHDK* genes, which converts $N_2$ to ammonia at ambient temperature and pressure, called biological nitrogen fixation (BNF)[11]. Fixed ammonia is subsequently assimilated into glutamate through the glutamine synthetase/glutamate:2-oxoglutarate aminotransferase (GS-GOGAT) pathway, and the glutamate thus formed is then used for cell growth and the renewal of nitrogenase to maintain diazotrophy[11,12]. Nitrogen fixation is a highly energy-consuming cellular process because it requires a large amount of adenosine triphosphate (ATP) and highly reductive electrons derived from ferredoxin or flavodoxin[13]. Therefore, diazotrophs have a system that strongly inhibits

[1]Graduate School of Agricultural and Life Sciences, The University of Tokyo, Bunkyo-ku, Tokyo, Japan. [2]Kikkoman Corporation, Noda, Chiba, Japan. [3]Collaborative Research Institute for Innovative Microbiology, The University of Tokyo, Bunkyo-ku, Tokyo, Japan. [4]RIKEN Center for Sustainable Resource Science, Wako, Saitama, Japan. ✉e-mail: babubabu-babubu@g.ecc.u-tokyo.ac.jp; umanis@g.ecc.u-tokyo.ac.jp

nitrogen fixation in the presence of extracellular inorganic/organic nitrogen-containing compounds, such as ammonium, glutamine, and asparagine, to avoid energy loss[14]. In addition, nitrogenases contain several metal clusters that are highly sensitive to oxygen[15]. In *Klebsiella oxytoca*, a well-studied diazotroph, NifA and NifL regulatory proteins control the expression of nitrogen fixation-related genes (*nif* genes) in response to oxygen and the availability of nitrogen by sensing the intracellular redox state and intracellular 2-oxoglutarate (2-OG)-to-glutamine ratio, respectively[16]. These regulatory systems limit culture conditions with carbon, nitrogen, and oxygen to activate diazotrophy, making it difficult to develop a bioproduction system for nitrogen-containing compounds using diazotrophs. Although BNF has been applied in biofertilization to promote plant growth[17], few studies have reported bioproduction using BNF. Although diazotrophic *Zymomonas mobilis* and *Rhodopseudomonas palustris* have been shown to produce BNF-based ethanol and poly(3-hydroxybutyrate), respectively, under nitrogen-fixing conditions[18–20], these are not nitrogen-containing compounds and nitrogen fixation was only used to support bacterial growth. The production of riboflavin, a nitrogen-containing heterocyclic compound, by diazotrophically grown *Xanthobacter autotrophicus* was recently reported[21], with *X. autotrophicus* being engineered to overexpress the genes involved in riboflavin biosynthesis;

however, the production titer (1–2 μg L$^{-1}$) was markedly lower than industrial bioproduction levels of ~10 g L$^{-1}$ using *Bacillus subtilis*[22].

In the present study, we attempted to establish a sustainable and eco-friendly method for the production of glutamate and other nitrogen-containing compounds via BNF, which uses abundant aerial nitrogen as a nitrogen source. We selected *K. oxytoca* NG13 (NG13), originally isolated from a rice rhizosphere in Japan[23], as a diazotrophic host to produce glutamate from aerial nitrogen for the following reasons: (1) ATP and reducing power for nitrogen fixation are provided by glycolysis in *K. oxytoca*[24], which is different from other diazotrophs such as *Azotobacter* or *Azospirillum* species that depend on the tricarboxylic acid (TCA) cycle to obtain energy components for nitrogen fixation[25,26]. This is a key point for glutamate production from aerial nitrogen because glutamate is synthesized from 2-OG, a compound composing the TCA cycle, and, thus, the enhancement of carbon flux from 2-OG to glutamate generally competes with the TCA cycle to obtain energy for nitrogen fixation. *K. oxytoca*, which fixes nitrogen independent of the TCA cycle, is not expected to cause metabolic conflict and is compatible with glutamate production. (2) A genetic manipulation system has been established for this strain[27]. The engineering design in the present study is outlined in Fig. 1. Three critical strategies considered are: (1) carbon sources in medium to enhance the nitrogenase activity of NG13, (2)

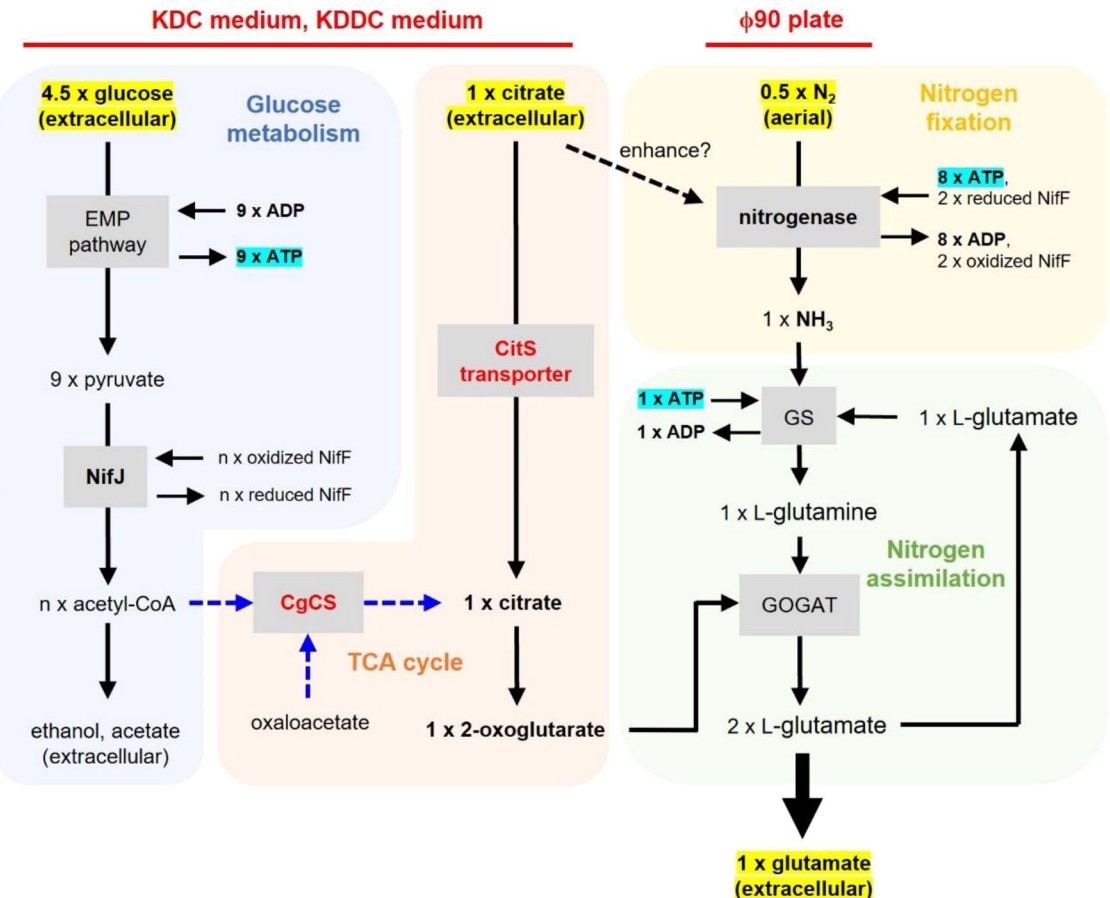

**Fig. 1 | Outline of engineering designs in the present study representing metabolism relevant to glutamate production from aerial nitrogen using *K. oxytoca*.** Engineering designs in this study were emphasized with red color. Klebsiella Dextrose Citrate medium (KDC medium) containing 7.5 g L$^{-1}$ each of glucose and citrate was used to enhance the nitrogenase activity of *K. oxytoca* NG13. Citrate synthase from *C. glutamicum* (CgCS) and the Na$^+$-dependent citrate symporter from NG13 (CitS) were overproduced to increase the carbon flux to 2-OG, a direct precursor of glutamate. φ90 plates were used to increase the aerial nitrogen supply to cells and thereby improve the glutamate production titer and rate. Klebsiella Doubled-Dextrose Citrate (KDDC) medium containing 15 g L$^{-1}$ glucose and 7.5 g L$^{-1}$ citrate extended the diazotrophic

period. Glucose is catabolized by the EMP pathway to produce ATP, and the NifJ-dependent conversion of pyruvate to acetyl-CoA provides electrons to NifF (flavodoxin), which transfers electrons to nitrogenase. The ATP produced is utilized for nitrogen assimilation by the GS-GOGAT pathway. In consideration of ATP production and consumption (highlighted in blue), one mole of glutamate is theoretically produced from 4.5 moles of glucose. Stoichiometry catalyzed by NifJ is expressed by n because other metabolic pathways branching from pyruvate are expected to be active. In the strain overproducing both CgCS and CitS (CgCS+CitS strain), extracellularly produced glutamate originated from citrate only and the carbon flow of CgCS (dotted blue arrow) did not appear to be active.

**Fig. 2 | Extracellular glutamate production by CgCS strain of *K. oxytoca* in φ18 test tubes.** *K. oxytoca* NG13 (black) and the strain overproducing citrate synthase from *C. glutamicum* (CgCS strain, red) were cultured statically under air in non-hermetic φ18 test tubes containing 5 mL of KD medium (dotted) or KDC medium (solid). Nitrogenase activity (**a**), cell number (**b**), glucose consumption (**c**), and extracellular glutamate production (**d**) were measured at each time point using independent triple biological replicates. Most of the data points of CgCS (KD) and CgCS (KDC) in (**b**) and NG13 (KD) and NG13 (KDC) in (**d**) overlapped. Data are shown as the mean of three biological replicates with standard deviation.

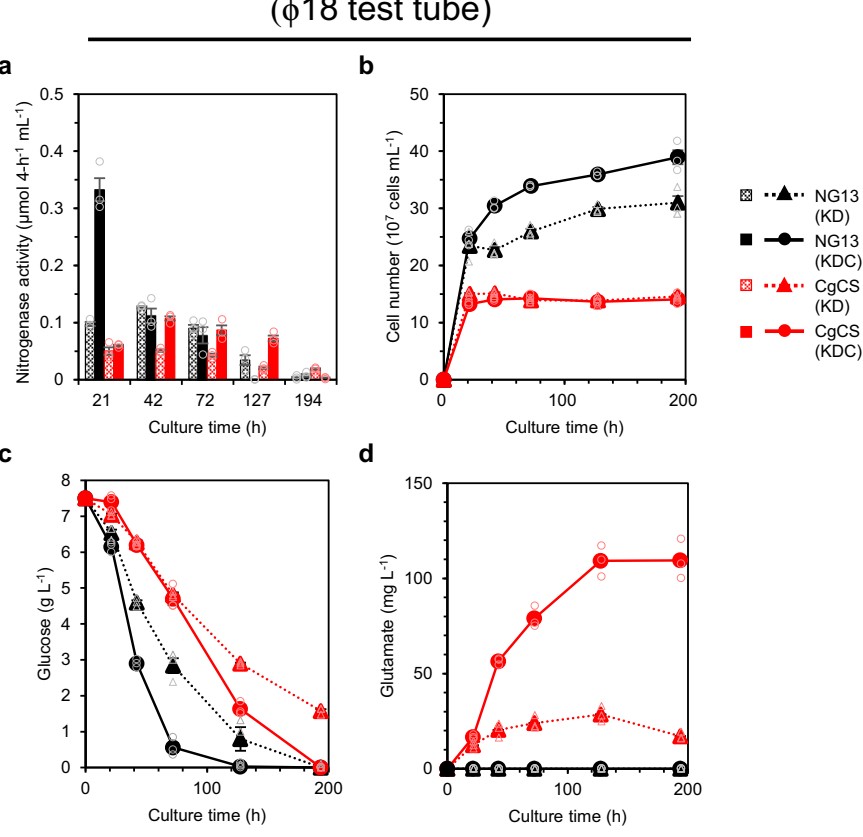

## Results

### Glucose and citrate as carbon sources for enhanced nitrogenase activity

To evaluate basal culture conditions for diazotrophic NG13 growth, a small amount (100 mg L$^{-1}$) of yeast extract was added to the culture medium to promote initial growth and shorten the lag time preceding nitrogenase activity. After a 26-h culture under these conditions, 0.2 µmol 4-h$^{-1}$ mL$^{-1}$ nitrogenase activity was detected (Supplementary Fig. 1). In the absence of yeast extract, NG13 growth decreased to less than one-twentieth and nitrogenase activity was not detected during a 5-day culture (Supplementary Fig. 1). Although only limited nitrogenase activity was observed in the presence of 100 mg L$^{-1}$ NH$_4$Cl or 500 mg L$^{-1}$ glutamine, similar nitrogenase activity of 0.2 µmol 4-h$^{-1}$ mL$^{-1}$ was retained even in the presence of 7.5 g L$^{-1}$ glutamate after a 25-h culture (Supplementary Fig. 1). This feature is preferable for glutamate production because the accumulated glutamate does not inhibit diazotrophy. Since the nitrogenase activity of *Klebsiella* species is known to be tolerant of oxygen to some extent[28], NG13 was cultured microaerobically in medium containing 100 mg L$^{-1}$ yeast extract in a non-hermetic tube without shaking.

Carbon sources for enhanced nitrogenase activity were examined because they are directly utilized for ATP synthesis and electron supply during nitrogen fixation. In *K. oxytoca*, NifJ (pyruvate:-flavodoxin oxidoreductase) catalyzes the conversion of pyruvate to acetyl-CoA during glycolysis with the concomitant reduction of NifF (flavodoxin), and the reduced NifF then serves as an electron donor for the nitrogenase reaction[24]. Therefore, the nitrogenase activity of *Klebsiella* species has been measured in media containing glucose as

the sole carbon source[27,29]. We herein demonstrated that NG13 exhibited nitrogenase activity in Klebsiella Dextrose (KD) medium containing 7.5 g L$^{-1}$ glucose (Fig. 2a). Although nitrogenase activity was observed until the depletion of glucose in KD medium (Fig. 2c), doubling the glucose concentration to 15 g L$^{-1}$ did not increase nitrogenase activity (Supplementary Fig. 2). No or weak nitrogenase activity was observed in the presence of only an organic acid (Supplementary Fig. 3); however, a combination of glucose and citrate enhanced nitrogenase activity during the early culture stage (Fig. 2a). A combination of malate, succinate, and 2-OG with glucose also increased nitrogenase activity; however, the highest nitrogenase activity was achieved with a combination of glucose and citrate (Supplementary Fig. 2). Although enhanced nitrogenase activity correlated with an increase in the concentration of citrate (up to 15 g L$^{-1}$), a concentration of citrate >7.5 g L$^{-1}$ did not markedly enhance nitrogenase activity (Supplementary Fig. 4). Therefore, 7.5 g L$^{-1}$ glucose and 7.5 g L$^{-1}$ citrate was selected as the basic carbon sources in subsequent experiments, and this medium was named Klebsiella Dextrose Citrate (KDC) medium.

In KDC medium, NG13 growth exhibited two phases after initial growth promoted by yeast extract: a glucose-consuming phase and a citrate-consuming phase. During the glucose-consuming phase, NG13 exhibited diazotrophic growth with high nitrogenase activity without extracellular glutamate production (Fig. 2; Supplementary Fig. 5). The glucose consumption rate of NG13 was higher in KDC medium than in KD medium, whereas limited citrate was consumed during the glucose-consuming phase (Fig. 2c; Supplementary Fig. 5). Following glucose depletion, NG13 entered the citrate-consuming phase, exhibiting diazotrophic growth (Supplementary Fig. 5). Some *Klebsiella* species possess an anaerobic citrate-utilization gene cluster, which is inactive under glucose-rich conditions and activated under anaerobic conditions to utilize citrate for growth[30]. NG13 growth in KDC medium suggests that NG13 possesses this gene cluster.

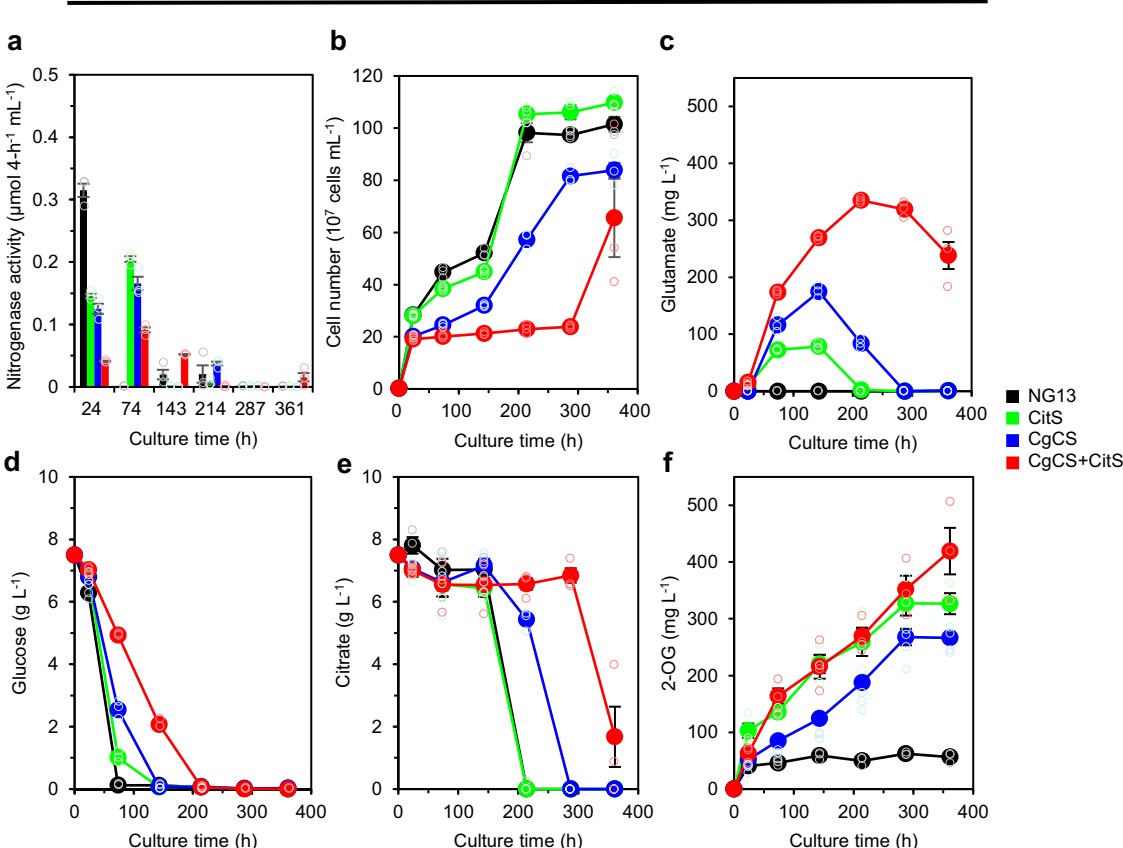

**Fig. 3 | Extracellular glutamate production by CitS and CgCS+CitS strains of *K. oxytoca* in φ25 test tubes.** *K. oxytoca* NG13 (black), CitS (green), CgCS (blue), and CgCS+CitS (red) strains were cultured statically under air in non-hermetic φ25 test tubes containing 5 mL of KDC medium. Nitrogenase activity (**a**), cell number (**b**), extracellular glutamate production (**c**), glucose consumption (**d**), citrate consumption (**e**), and 2-OG production (**f**) were measured at each time point using independent triple biological replicates. Data are shown as the mean of three biological replicates with standard deviation.

## Extracellular glutamate production by heterologous citrate synthase under diazotrophic conditions

Glutamate fermentation using *C. glutamicum* is currently conducted aerobically by supplying a large amount of glucose and ammonium. Glucose is metabolized to 2-OG via glycolysis and the TCA cycle, while ammonium is incorporated into 2-OG to produce glutamate mainly by glutamate dehydrogenase[31]. Under diazotrophic conditions requiring low oxygen concentrations, the TCA cycle is less active because of regulation by the ArcAB two-component system in many facultative anaerobes, including *K. oxytoca*[32,33], which may be a limiting factor for extracellular glutamate production. To increase carbon flux to the TCA cycle, we focused on citrate synthase (CS). CS catalyzes the initial reaction in the TCA cycle, a Claisen condensation-like reaction that uses acetyl-CoA and oxaloacetate (OAA) to generate citrate and CoA. There are two types of CSs: dimeric type I CS and hexameric type II CS[34,35]. Type I CSs are found in eukaryotes, archaea, and most gram-positive bacteria, whereas type II CSs are found in most gram-negative bacteria, including *K. oxytoca*[36]. While type II CSs are allosterically regulated by NADH, type I CSs are not[36]. The introduction of the gene encoding NADH-insensitive type I CS from *C. glutamicum* (CgCS) into non-diazotrophic *Pantoea ananatis* induced extracellular glutamate production from supplied ammonium[31]. Therefore, we introduced a pCCS plasmid, which expressed the *CgCS* gene under the control of the *trc* promoter, into *K. oxytoca* NG13 to yield a transformant named the CgCS strain. Culturing the strain in a φ18 test tube with 5 mL KD medium produced 28 mg L$^{-1}$ extracellular glutamate after 127 h (Fig. 2d) with a production rate of 0.22 mg L$^{-1}$ h$^{-1}$. During the culture, the cell growth of the CgCS strain

stopped after 20 h of cultivation and was kept low, at ~50% that of the wild-type (Fig. 2b), which suggested that glutamate production by the CgCS strain was achieved at the expense of cell growth. A decrease in nitrogenase activity was observed for the CgCS strain in the KD medium culture (Fig. 2a). Since this reduction was attributed to a low cell number, the CgCS strain was cultured in KDC medium to enhance the nitrogenase activity. In KDC medium, the nitrogenase activity of the CgCS strain was restored, and 109 mg L$^{-1}$ glutamate was produced after 127 h (Fig. 2a, d) with a production rate of 0.88 mg L$^{-1}$ h$^{-1}$. Glutamate production by the CgCS strain correlated with the concentration of citrate in the medium (Supplementary Fig. 4). The glucose consumption rate of the CgCS strain was increased in KDC medium (Fig. 2c), potentially resulting in the production of more glutamate than in KD medium.

2-Methylcitrate synthase (MCS), which catalyzes the condensation of propionyl-CoA and OAA to yield 2-methylcitrate and CoA, can also catalyze the CS reaction without NADH inhibition[37]. The expression of the gene encoding MCS from *C. glutamicum* (*CgMCS* gene) or *Escherichia coli* (*EcMCS* gene) in NG13, introduced by pCMCS or pEMCS plasmids, respectively, provided similar glutamate production levels to that with the CgCS strain (Supplementary Fig. 6). The overproduction of a putative NADH-sensitive type II CS from NG13 (KoCS) also resulted in glutamate production but at a markedly lower level than that with CgCS or MCSs (Supplementary Fig. 6). A high NADH-to-NAD$^+$ ratio has been reported when *E. coli* is grown on glucose under anaerobic conditions[38]. These results suggest that NADH-insensitive CS was effective for glutamate production even under microaerobic conditions.

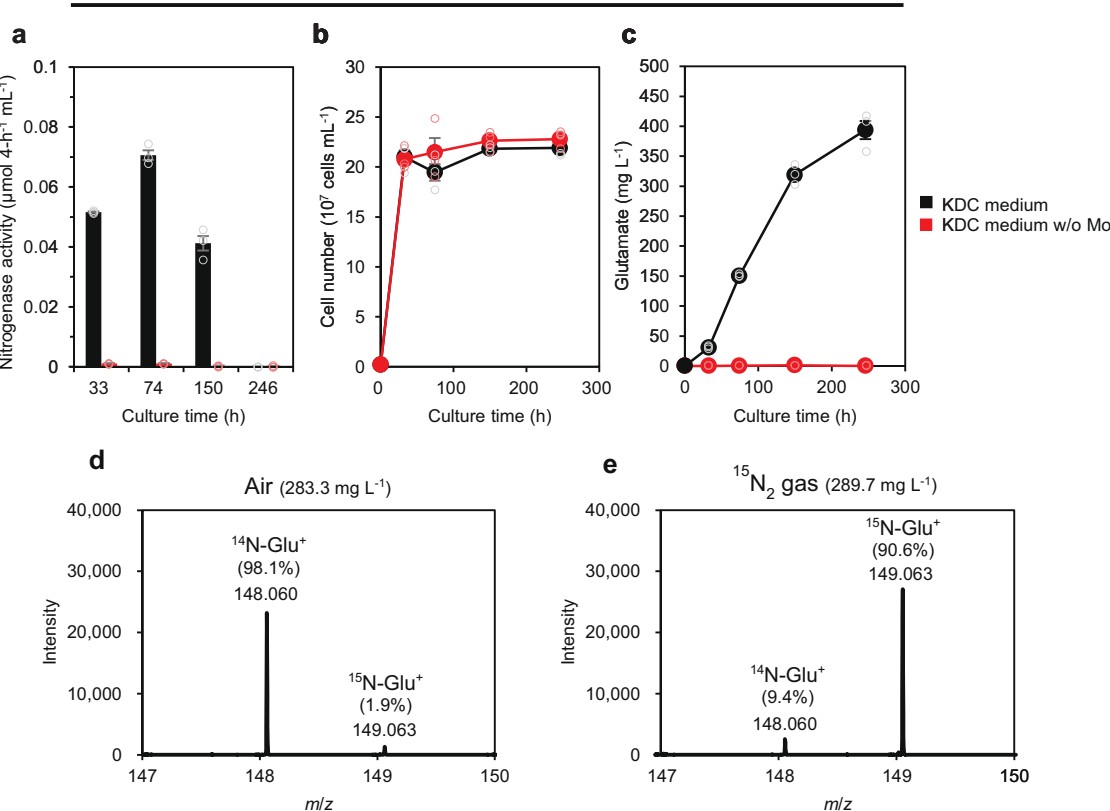

**Fig. 4 | Extracellularly produced glutamate was derived from aerial nitrogen.** The CgCS+CitS strain was cultured statically under air in non-hermetic ϕ25 test tubes containing 5 mL of KDC medium (black) or KDC-based medium without molybdenum (red). Nitrogenase activity (**a**), cell number (**b**), and extracellular glutamate production (**c**) were measured at each time point using independent triple biological replicates. Data are shown as the mean of three biological replicates with standard deviation. The results of LC–MS analyses of extracellularly produced glutamate by the CgCS+CitS strain are shown in (**d**) and (**e**). The CgCS+CitS strain was cultured statically in hermetic ϕ25 test tubes containing 5 mL of KDC medium under conditions with air (**d**) or $^{15}N_2$ gas (**e**). The concentrations of extracellularly produced glutamate, 283.3 mg $L^{-1}$ and 289.7 mg $L^{-1}$, are shown in parenthesis in (**d**) and (**e**), respectively. The theoretical mass/charge (*m/z*) values of positively charged $^{14}N$- and $^{15}N$-glutamate are 148.060 and 149.063, respectively. Ratio (%) represents the ratio of the intensity and the total intensity of $^{14}N$-Glu$^+$ and $^{15}N$-Glu$^+$.

## Effects of air availability on glutamate production

The effects of air availability on glutamate production were evaluated by changing the culture volume. Since NG13 is not motile, NG13 cells sink to the bottom of the test tube in a static culture and, thus, a reduced culture volume was expected to increase air availability. A reduction in the culture volume by decreasing KDC medium from 5 mL (a height of 3.0 cm in a ϕ18 test tube) to 2.5 mL (a height of 1.5 cm in a ϕ18 test tube) increased glutamate production by the CgCS strain to 157 mg $L^{-1}$ with a production rate of 1.25 mg $L^{-1}$ $h^{-1}$, whereas increasing the KDC medium to 10 mL (a height of 5.4 cm in a ϕ18 test tube) decreased glutamate production to 61 mg $L^{-1}$ with a rate of 0.81 mg $L^{-1}$ $h^{-1}$ (Supplementary Fig. 7). Similarly, culturing the CgCS strain in 5 mL KDC medium in a ϕ25 test tube (a height of 1.4 cm) increased glutamate production to 175 mg $L^{-1}$ with a production rate of 1.22 mg $L^{-1}$ $h^{-1}$ (Figs. 2d and 3c). These results imply that aerial nitrogen and/or oxygen availability may be a critical factor for enhancing glutamate production. Therefore, we used ϕ25 test tubes with higher glutamate production for the subsequent cultures.

## Effects of the overproduction of a citrate transporter in the CgCS strain

The addition of TCA cycle components, such as malate, succinate, and 2-OG, to the KD medium enhanced the nitrogenase activity of the CgCS strain. However, glutamate production was not as high as that obtained using citrate (Supplementary Fig. 2), suggesting that citrate is a key compound required for enhanced glutamate production. When the CgCS strain

was cultured in 5 mL of KDC medium in a ϕ25 test tube, it exhibited high nitrogenase activity and produced glutamate but did not consume citrate during the glucose-consuming phase (Fig. 3). This may be attributed to the low expression of the *citS* gene, which encodes a Na$^+$-dependent citrate symporter and is located in the anaerobic citrate-utilization gene cluster described above, during this phase[39]. Therefore, we cloned and overexpressed the *citS* gene from NG13 under the control of the *trc* promoter on the pCDF-derived plasmid to increase extracellular citrate uptake. When only *citS* was overexpressed in NG13 (CitS strain), glutamate production of 78 mg $L^{-1}$ was observed in KDC medium at 143 h with a production rate of 0.55 mg $L^{-1}$ $h^{-1}$, which was lower than that with the CgCS strain under the same culture conditions (Fig. 3c). However, when *citS* was overexpressed in the CgCS strain (CgCS+CitS strain), glutamate production reached 335 mg $L^{-1}$ (2.28 mM) with a rate of 1.57 mg $L^{-1}$ $h^{-1}$ (Fig. 3c). A similar seemingly additive effects of CgCS and CitS were observed in 5 mL culture of KDC medium in a ϕ18 test tube (Supplementary Fig. 8). The increased accumulation of 2-OG was observed in all glutamate-producing strains (Fig. 3f), suggesting that carbon flux to 2-OG is a rate-limiting step for glutamate production, which may be increased by the expression of CgCS and/or CitS. Despite differences in glutamate production, the CgCS, CitS, and CgCS+CitS strains exhibited two-phase growth, similar to NG13. During the glucose-consuming phase, all the strains exhibited high nitrogenase activity accompanied by extracellular glutamate production (Fig. 3a, c, d). No marked differences were observed in citrate consumption between *citS*-expressing strains (the CitS and CgCS+CitS strains) and others (the

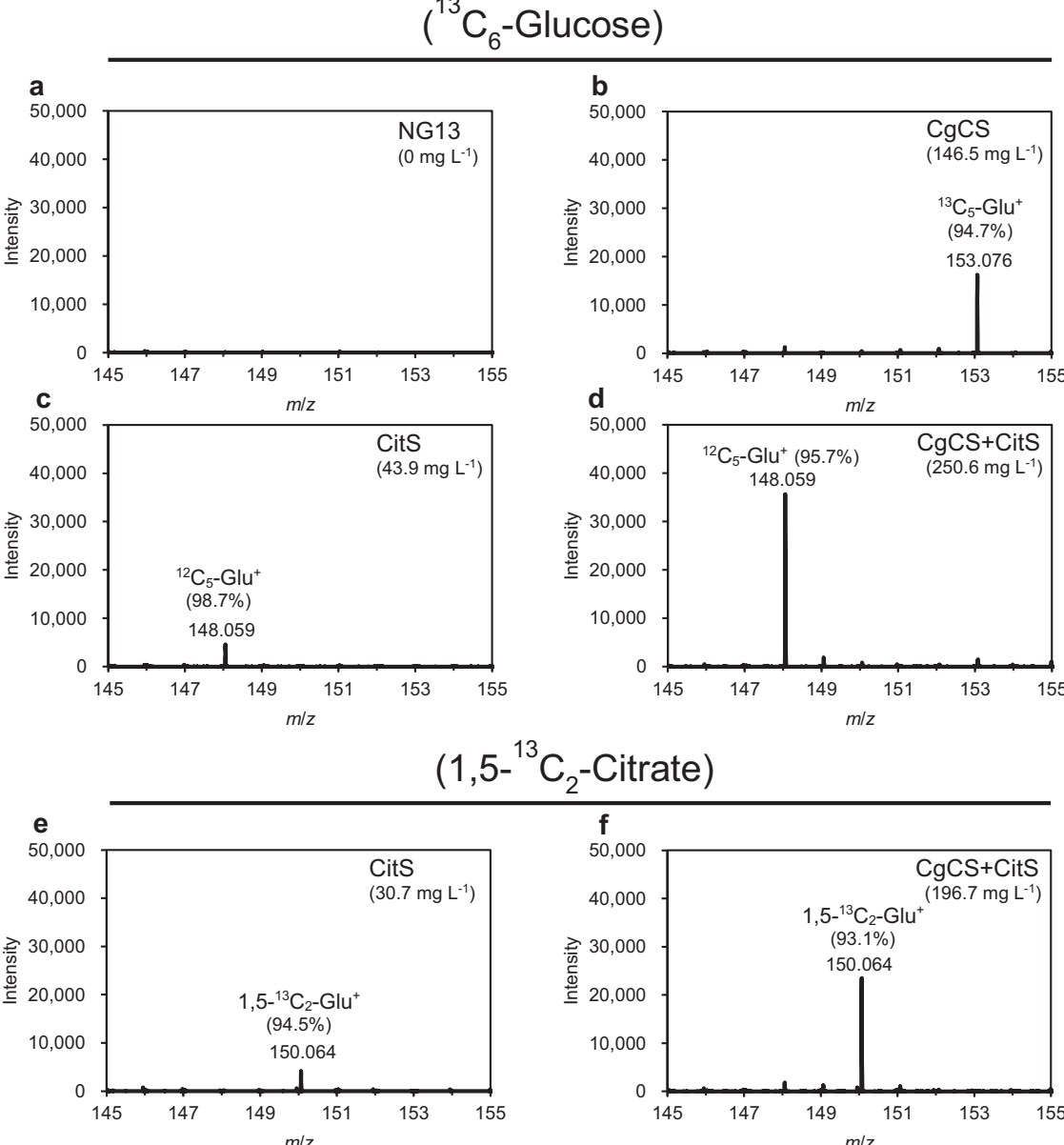

**Fig. 5 | Carbon origin of extracellular glutamate.** *K. oxytoca* NG13 (**a**), CgCS (**b**), CitS (**c**), and CgCS+CitS (**d**) strains were cultured statically under air in non-hermetic ϕ25 test tubes containing 5 mL of KDC medium composed of labeled $^{13}C_6$-glucose and normal $^{12}C$-citrate, and the mass of glutamate produced in the supernatants were analyzed by LC–MS. $^{13}C_6$-glucose is metabolized to $^{13}C_5$-glutamate. The theoretical *m/z* values of positively charged normal $^{12}C_5$- and $^{13}C_5$-glutamate are 148.059 and 153.076, respectively. Ratio (%) represents the ratio of the intensity and the total intensity of $^{12}C_5$-Glu$^+$ and $^{13}C_5$-Glu$^+$. CitS (**e**) and CgCS+CitS (**f**) strains were cultured statically under air in KDC medium composed of normal $^{12}C$-glucose and 1,5-$^{13}C_2$-citrate, and the concentration of glutamate produced in the supernatants were analyzed by LC–MS. 1,5-$^{13}C_2$-citrate is metabolized to 1,5-$^{13}C_2$-glutamate. The theoretical *m/z* value of positively charged 1,5-$^{13}C_2$-glutamate is 150.064. Ratio (%) represents the ratio of the intensity and the total intensity of $^{12}C_5$-Glu$^+$ and 1,5-$^{13}C_2$-Glu$^+$. Concentrations of extracellularly produced glutamate were provided in the parenthesis.

wild-type NG13 and CgCS strain) during the glucose-consuming phase in ϕ25-tube cultures (Fig. 3e); however, this may be due to low quantification sensitivity. After glucose depletion, the citrate-consuming phase commenced (Fig. 3d, e), during which extracellular glutamate production decreased and the second cell growth started (Fig. 3b, c). Nitrogenase activity during the citrate-consuming phase was markedly lower than that during the glucose-consuming phase (Fig. 3a), suggesting that some of the glutamate synthesized was used as a nitrogen source for cell growth.

**Aerial N₂ for glutamate production**
Two different experiments were performed to confirm that the extracellular glutamate produced by the CgCS+CitS strain was derived from

aerial nitrogen. Molybdenum is essential for nitrogenase activity because it is a component of the FeMo-cofactor forming the nitrogenase active site[40]. In a culture of the CgCS+CitS strain in a KDC-based medium without molybdenum, neither nitrogenase activity nor extracellular glutamate production was observed, while cell growth remained unaffected (Fig. 4a–c). This result suggests that the glutamate produced was synthesized via nitrogen fixation. To further verify this, the CgCS+CitS strain was cultured in KDC medium with $^{15}N_2$-containing gas, and the metabolites in the culture supernatant were analyzed using liquid chromatography–mass spectrometry (LC–MS). When the CgCS strain was cultured under normal conditions in air, glutamate was detected at *m/z* 148.060 (98.1%) and 149.063 (1.9%), which represent $^{14}N$- and

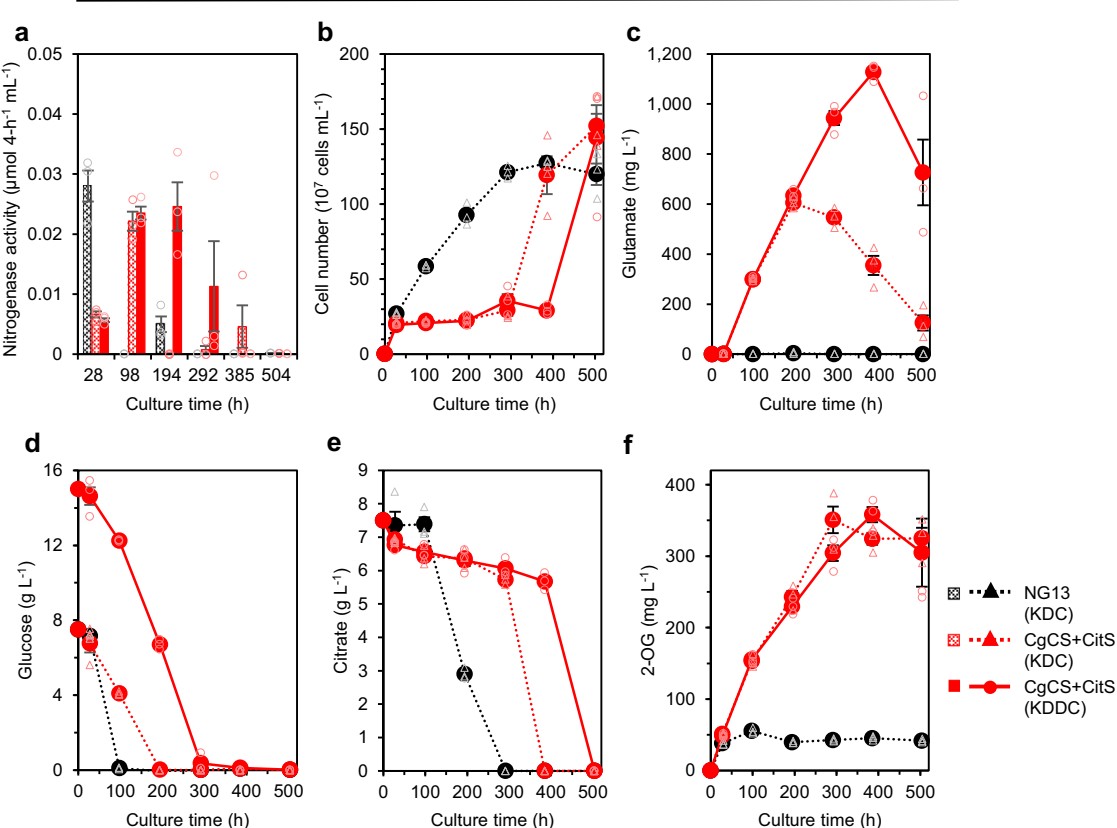

**Fig. 6 | Glutamate production by the CgCS+CitS strain of *K. oxytoca* in ϕ90 plate culture.** *K. oxytoca* NG13 (black) and CgCS+CitS strain (red) were cultured statically under air in non-hermetic ϕ90 plates containing 25 mL of KDC (dotted) and glucose-doubling KDDC (solid) medium, respectively. Nitrogenase activity (**a**), cell number (**b**), extracellular glutamate production (**c**), glucose consumption (**d**), citrate consumption (**e**), and 2-OG production (**f**) were measured at each time point using independent triple biological replicates. Data points of CgCS+CitS cultured in KDC and KDDC media, which are indicated as CgCS+CitS (KDC) and CgCS+CitS (KDDC), overlapped until 200 h in (**b**), (**c**), (**e**), and (**f**). Data are shown as the mean of three biological replicates with standard deviation.

[15]N-containing glutamate, respectively (Fig. 4d). In contrast, under conditions in which air was replaced with [15]N$_2$-containing gas, a major peak was detected at *m/z* 149.063 (90.6%) (Fig. 4e), which indicates that glutamate produced by the CgCS+CitS strain was derived from aerial nitrogen.

**Carbon origin for glutamate production**

To identify the origin of the carbon of glutamate produced in the KDC medium culture, incorporation assays using labeled carbon substrates were performed. Wild-type NG13, CgCS, CitS, and CgCS+CitS strains were cultured in modified KDC medium containing [13]C$_6$-glucose. LC–MS analyses of culture supernatants confirmed the lack of glutamate production by intact NG13 (Fig. 5a), whereas the CgCS strain produced glutamate at 153.076 *m/z*, indicating that this strain synthesized glutamate from labeled glucose (Fig. 5b). In contrast, the CitS strain produced non-labeled glutamate at 148.059 *m/z*, suggesting that this strain synthesized glutamate from citrate (Fig. 5c). Unexpectedly, the CgCS+CitS strain primarily produced glutamate at 148.059 *m/z*, indicating that most of the glutamate produced originated from citrate (Fig. 5d). To confirm this, the CitS and CgCS+CitS strains were cultured in modified KDC medium containing 1,5-[13]C$_2$-citrate. LC–MS analyses revealed that both strains produced glutamate mainly at 150.064 *m/z*, which suggested that the glutamate produced originated from citrate in both strains (Fig. 5e, f). These results indicate that carbon flow for glutamate synthesis in the CgCS+CitS strain markedly differed from that in the CgCS strain.

**Improved culture conditions for glutamate production exceeding 1 g L$^{-1}$**

To further increase air availability and thereby enhance glutamate production, we used 25 mL KDC medium in a ϕ90 plate (a height of 0.6 cm). Under these conditions, the CgCS+CitS strain diazotrophically produced up to 619 mg L$^{-1}$ (4.11 mM) glutamate by day 9 with a production rate of 3.11 mg L$^{-1}$ h$^{-1}$ (Fig. 6c, red dotted line), which markedly exceeded 335 mg L$^{-1}$ in a ϕ25 test tube with a rate of 1.57 mg L$^{-1}$ h$^{-1}$ (Fig. 3c, red solid line). While NG13 consumed limited citrate during the glucose-consuming phase, the CgCS+CitS strain consumed 1.14 g L$^{-1}$ (5.92 mM) citrate in KDC medium by day 9 (Fig. 6e). Under these conditions, CgCS+CitS cells settled on the bottom of the plate, forming biofilm-like aggregates, which were rarely observed in the ϕ18 or ϕ25 test tubes. During glutamate production, the cell number of the CgCS+CitS strain remained constant at ~2 × 10[8] cells mL$^{-1}$, which corresponded to OD$_{660}$ = 0.29 (Fig. 6b). The accumulation of 2-OG by the CgCS+CitS strain was not greater in the ϕ90 plate than in the ϕ25 test tube (Figs. 3f and 6f), suggesting that increased oxygen availability, which may activate the TCA cycle to produce 2-OG, was not the main factor contributing to enhanced glutamate production in the ϕ90 plate. Therefore, we examined aerial nitrogen availability as an alternative explanation for increased glutamate production in the ϕ90 plate. When the CgCS+CitS strain was cultured under non-diazotrophic conditions supplied with 1 g L$^{-1}$ NH$_4$Cl, the glutamate production profile was similar in the ϕ25 test tube and ϕ90 plate (Supplementary Fig. 9). These results suggest that

**Table 1 | Titer, production rate, and productivity of extracellular glutamate of engineered *K. oxytoca* strains in this study**

| Culture | Strain | Time maximum glutamate produced (h) | Cell number ($10^7$ cells mL$^{-1}$) | Citrate consumption (mM) | Glutamate production (mg L$^{-1}$) [mM] | Glutamate production rate (mg L$^{-1}$ h$^{-1}$) | Citrate-based glutamate productivity[a] (%) | Glucose-based glutamate productivity[b] (%) |
|---|---|---|---|---|---|---|---|---|
| ϕ25-KDC | NG13[c] | 74 | 44.7 ± 1.0 | 2.49 ± 1.86 | 0 [0] | 0 | 0 | 0 |
| | CitS | 143 | 44.9 ± 0.5 | 5.52 ± 0.90 | 78.5 ± 2.42 [0.53 ± 0.02] | 0.55 ± 0.02 | 10.7 ± 2.1 | 5.8 ± 0.2 |
| | CgCS +CitS | 214 | 22.9 ± 0.8 | 4.85 ± 0.95 | 335 ± 2.16 [2.28 ± 0.02] | 1.57 ± 0.01 | 51.7 ± 8.0 | 24.6 ± 0.2 |
| ϕ90-KDC | CgCS +CitS | 194 | 22.8 ± 1.6 | 5.93 ± 0.68 | 605 ± 10.6 [4.11 ± 0.07] | 3.11 ± 0.05 | 71.7 ± 7.0 | 44.5 ± 0.8 |
| ϕ90-KDDC | CgCS +CitS | 385 | 29.1 ± 1.5 | 9.51 ± 0.63 | 1128 ± 16.2 [7.67 ± 0.11] | 2.93 ± 0.04 | 81.7 ± 5.6 | 41.4 ± 0.6 |

Data are shown as the mean of three biological replicates with standard deviation.

[a]100% indicates that all consumed citrate was converted to glutamate.

[b]100% indicates that all ATP produced from glucose by the EMP pathway was consumed only by $N_2$-fixation and the GS/GOGAT pathway to produce glutamate.

[c]Data of NG13 are shown at the time when glucose was exhausted.

improved glutamate production in the ϕ90 plate under diazotrophic conditions is attributed to increased aerial nitrogen availability.

Since the glucose-consuming phase exhibited sufficient nitrogenase activity to allow active glutamate production, we hypothesized that an increase in the initial glucose content of the medium may extend the glucose-consuming phase, thereby sustaining nitrogenase activity to produce more glutamate. This hypothesis was confirmed by the observation that the use of Klebsiella Doubled-Dextrose Citrate (KDDC) medium containing 15 g L$^{-1}$ glucose and 7.5 g L$^{-1}$ citrate in ϕ90 plates extended the diazotrophic period to day 16 and glutamate production of the CgCS+CitS strain reached 1.13 g L$^{-1}$ (7.67 mM) on day 16 with a production rate of 2.93 mg L$^{-1}$ h$^{-1}$, accompanied by the concurrent consumption of 1.83 g L$^{-1}$ (9.51 mM) citrate (Fig. 6a, c, d, e, red solid line; Table 1).

## Discussion

We herein achieved BNF-based extracellular glutamate production using aerial nitrogen and optimized a system for glutamate production by three key strategies (Fig. 1). A combination of glucose and citrate as carbon sources was identified as a critical factor for extracellular glutamate production. Although citrate alone was not an optimum carbon source for diazotrophic NG13 growth (Supplementary Fig. 3), it enhanced nitrogenase activity and glutamate production by the CgCS strain in a dose-dependent manner when combined with glucose (Fig. 2a, d; Supplementary Fig. 4). Since limited citrate was consumed in the presence of glucose (Fig. 3d, e; Supplementary Fig. 5) and the glutamate produced by the CgCS strain was derived from glucose (Fig. 5b), citrate may increase glutamate production by enhancing nitrogenase activity; however, the underlying mechanisms require further investigation.

The second key strategy was the overproduction of CgCS and CitS. CgCS condenses glucose-derived acetyl-CoA with OAA to yield citrate, while CitS imports extracellular citrate into cells. Since citrate acquired via these two routes may contribute to the synthesis of glutamate, and CgCS and CitS appeared to additively contribute to glutamate production (Fig. 3c; Supplementary Fig. 8), the glutamate produced by the CgCS+CitS strain was expected to be derived from both glucose and citrate. However, incorporation experiments using labeled carbon substrates revealed that glutamate produced by the CgCS+CitS strain was only derived from extracellular citrate (Fig. 5d, f). Therefore, the seemingly additive effects of CgCS and CitS may not be simply explained by their cumulative positive effects on glutamate production. We hypothesized that carbon flow for glutamate synthesis was markedly affected in the CgCS+CitS strain, and that glutamate overproduction or enhanced citrate uptake induced by CitS may inhibit the conversion of glucose to glutamate. Carbon flux in the CgCS+CitS strain in the presence of citrate requires further investigation to elucidate the mechanism by which CgCS and CitS modulate glutamate production.

The third key strategy is to enhance the aerial nitrogen supply. An increased air availability resulted in the production of more glutamate by the CgCS+CitS strain (Figs. 3c and 6c; Supplementary Fig. 8). Although we initially assumed that oxygen may enhance glutamate production by activating the TCA cycle to increase the supply of 2-OG, greater air availability did not promote 2-OG production by the CgCS+CitS strain (Figs. 3f and 6f), which suggests that oxygen availability was not increased in these cells. Furthermore, under non-diazotrophic conditions supplied with 1 g L$^{-1}$ NH$_4$Cl, glutamate production in both ϕ25 test tubes and ϕ90 plates, and the production profiles of both cultures were similar (Supplementary Fig. 9). Therefore, an aerial nitrogen supply appears to be a rate-limiting step in glutamate production in the CgCS+CitS strain. Biofilm-like aggregates were observed in ϕ90-plate cultures, which may positively affect sustainable nitrogen fixation by protecting oxygen-sensitive nitrogenase from increased levels of oxygen. Although dissolved nitrogen and oxygen concentrations in the culture were not measured, appropriate aeration, particularly the balance between aerial nitrogen and oxygen supplies, will be required to promote the production of glutamate. To elucidate the relationship between biofilm formation and glutamate production, we plan to analyze biofilm formation and biofilm-related gene expression in the CgCS+CitS strain in future studies.

By implementing these three key strategies, we herein successfully improved the titer and rate of glutamate production from aerial nitrogen, and achieved glutamate production >1 g L$^{-1}$ in a ϕ90 plate in a KDDC medium culture (Fig. 6c; Table 1). We calculated the productivity of glutamate based on the consumption ratio of glucose to produce ATP for glutamate production, assuming that 4.5 moles of glucose are theoretically consumed to synthesize one mole of glutamate (see Methods section for details). Based on this assumption, 1.36 g L$^{-1}$ (9.24 mM) glutamate may be maximally produced in the KDC (containing 41.6 mM glucose) medium by the CgCS+CitS strain, which was used to evaluate glucose-based productivity. Glutamate production of 78 mg L$^{-1}$ (0.53 mM) and 335 mg L$^{-1}$ (2.28 mM) by the CitS and CgCS+CitS strains, respectively, under ϕ25-KDC conditions corresponded to 5.8 and 24.6%, respectively, in glucose-based productivity (Table 1). Productivity of the CgCS+CitS strain was further increased to 44.5% under ϕ90-KDC condition and 41.4% under ϕ90-KDDC condition. We also calculated glutamate productivity according to the conversion ratio from citrate to glutamate (citrate-based productivity) in the CitS and CgCS+CitS strains (Table 1). Citrate-based productivity increased from 10.7% under ϕ25-KDC condition by the CitS strain to 81.7% under ϕ90-KDDC condition by the CgCS+CitS strain.

Approximately 20 g L$^{-1}$ glutamate was produced by *C. glutamicum* using ammonium, while our method generated nearly 1 g L$^{-1}$, which was still low. This difference was attributed to the lower cell number corresponding to OD$_{660}$ of 0.29 in our 1 g L$^{-1}$ culture with a ϕ90 plate than that of *C. glutamicum* (around OD$_{660}$ = 40)[41]. When the glutamate production

per cell was compared between the *C. glutamicum* system and ours, ours ($3.8\,\mathrm{g\,L^{-1}\,OD_{660}^{-1}}$) surpassed that of *C. glutamicum* ($0.58\,\mathrm{g\,L^{-1}\,OD_{660}^{-1}}$). In terms of the production rate, the production of a nitrogen-containing compound at $\sim$3 mg L$^{-1}$ h$^{-1}$ was achieved in our culture system. Although our glutamate-producing rate was markedly lower than that of *C. glutamicum* (952 mg L$^{-1}$ h$^{-1}$), our fermentation process does not require an energy-consuming process, such as shaking or air supply. Future studies are required to investigate approaches to reduce the required culture time and increase cell number in the culture. Since amino acids and other nitrogen-containing compounds are derived from glutamate, they may be produced through BNF based on this sustainable and eco-friendly production process.

## Methods

### Bacterial strains and culture conditions

The diazotroph *K. oxytoca* NG13 (deposited at the National Institute of Technology and Evaluation, Japan, as NITE P-03721) was originally isolated from the rhizosphere of *Oryza sativa* C5444[23]. NG13 was routinely pre-cultured at 30 °C in LB liquid medium (1% Bacto Tryptone [Thermo Fisher Scientific Japan, Tokyo, Japan], 1% NaCl, and 0.5% Bacto Yeast Extract [Thermo Fisher Scientific Japan]) with vigorous shaking. Tetracycline (3 μg mL$^{-1}$) and spectinomycin (100 μg mL$^{-1}$) were used to select plasmid-harboring transformants. Prior to the diazotrophic culture, cells grown overnight in LB medium were harvested and resuspended in distilled water. In test tube cultures, $1 \times 10^{7}$ cells were inoculated into 5 mL liquid medium in φ18 × 180 mm or φ25 × 150 mm test tubes (AGC Techno Glass Co., Ltd., Shizuoka, Japan). In plate cultures, $5 \times 10^{7}$ cells were inoculated into 25 mL liquid medium in φ90 × 15 mm plates (AS ONE Co., Ltd., Osaka, Japan). Medium composition was based on Rennie medium[42], with some modifications, including 800 mg L$^{-1}$ K$_2$HPO$_4$, 200 mg L$^{-1}$ KH$_2$PO$_4$, 100 mg L$^{-1}$ NaCl, 98 mg L$^{-1}$ MgSO$_4$, 60 mg L$^{-1}$ CaCl$_2$·2H$_2$O, 1 mg L$^{-1}$ NaFe-EDTA, 1 mg L$^{-1}$ Na$_2$MoO$_4$·2H$_2$O, 10 mg L$^{-1}$ *p*-aminobenzoic acid, 5 mg L$^{-1}$ biotin, 100 mg L$^{-1}$ Bacto Yeast Extract, and appropriate carbon sources, specifically 7.5 g L$^{-1}$ glucose in KD medium; 7.5 g L$^{-1}$ each of glucose and citrate in KDC medium; and 15 g L$^{-1}$ glucose and 7.5 g L$^{-1}$ citrate in KDDC medium. To induce gene expression from the *trc* promoter in engineered *K. oxytoca* strains, 0.1 mM (φ18 test tubes) or 0.02 mM (φ25 test tubes and φ90 plates) isopropyl β-D-thiogalactopyranoside was added to the medium, at which glutamate production by the CgCS+CitS strain was the greatest. A static culture was routinely performed in test tubes covered with an aluminum cap at 25 °C statically. All chemicals without specified manufacturer's details were purchased from FUJIFILM Wako Pure Chemical Corporation (Osaka, Japan) and Sigma-Aldrich (St. Louis, MO, USA).

### Measurement of nitrogenase activity

Nitrogenase activity was measured as acetylene-reducing activity (ARA)[43]. At each relevant culture time point, test tubes were stopped with butyl rubbers (W-18 and W-24T for φ18 and φ25 test tubes, respectively) (Taiyo Kogyo Co., Ltd., Tokyo, Japan), and 15% of the gas phase was replaced with dissolved acetylene (Gas-Ken Co., Ltd., Tokyo, Japan). Tubes containing acetylene were incubated at 25 °C for 4 h. Test tubes were then thoroughly vortexed, and a 0.5-mL aliquot of the gas phase was analyzed using the gas chromatograph GC-8AIF equipped with a flame ionization detector (Shimadzu, Kyoto, Japan) and Porapak-T 80/100 column (GL Science, Tokyo, Japan). Analysis conditions were as follows: carrier gas, nitrogen gas; column temperature, 100 °C; and injection temperature, 120 °C. The amount of ethylene was calculated based on the detected peak areas, which were defined as ARA. In plate cultures, one-fifth (5 mL) of the whole culture (25 mL) was transferred into a φ18 test tube, which was cap-closed for nitrogenase activity measurements as described above. Special care was taken while pipetting the culture from the plate to the test tube to prevent nitrogenase damage as a result of oxygen exposure. The ethylene content in the gas phase of a tube containing only fresh medium without bacterial cells was used as a blank.

### Measurement of cell number and metabolites

After thorough vertexing or pipetting of the cells settled on the bottom of the test tube or plate, the total number of cells in the culture was measured using a Multisizer 3 Coulter counter equipped with a 30-μm aperture tube (Beckman Coulter, Brea, CA, USA). To measure metabolites, the culture was centrifuged at $20{,}000 \times g$ for 3 min to remove bacterial cells, and the metabolites in the supernatant were assayed using the L-glutamate kit "Yamasa" NEO (Yamasa Corporation, Chiba, Japan) and F-kit (D-glucose) (Roche/R-Biopharm, Darmstadt, Germany) in accordance with kit instructions. To quantify citrate and 2-OG, the culture supernatant was diluted 10-fold with methanol and transferred to a LC-2050C LT system (Shimadzu) connected to a LCMS-2050 system (Shimadzu). Samples were separated using a CAPCELL PAK C18 IF S2 column, 2 μm, 2.0 × 50 mm (OSAKA SODA, Osaka, Japan). LC conditions were as follows: mobile phase A, H$_2$O + 0.1% formic acid; mobile phase B, methanol + 0.1% formic acid; 5% B for 1 min, 5%–95% B over 6 min, 95% B for 4 min, 95%–5% B over 1 min, and 5% B for 4 min, at a flow rate of 0.3 mL min$^{-1}$. MS analyses were simultaneously performed using electrospray ionization in the negative mode. Ions of *m/z* 191.02 in 1.2 ± 0.2 min and *m/z* 145.01 in 1.4 ± 0.2 min were analyzed to quantify citrate and 2-OG, respectively.

### Plasmid construction

The plasmids used in the present study are listed in Supplementary Table 1. These plasmids were constructed by the Gibson assembly method using DNA fragments amplified through PCR with the primer sets listed in Supplementary Table 2. BBD29_04605, encoding CgCS, and BBD29_03720, encoding CgMCS, were cloned from the *C. glutamicum* ATCC13869 genome (GenBank: CP016335.1). The *prpC* gene, encoding EcMCS, was cloned from the *E. coli* W3110 genome (GenBank: AP009048.1). Genes encoding putative CS (KoCS, Supplementary Fig. 10) and a Na$^+$-dependent citrate symporter (CitS, Supplementary Fig. 11) were cloned from the *K. oxytoca* NG13 genome based on the genome sequence information of *Klebsiella* sp. M5a1 (GenBank: CP020657.1). The genes encoding CgCS, CgMCS, EcMCS, and putative KoCS were expressed in pSC101-derived plasmids under the control of the *trc* promoter, and were designated as pCCS, pCMCS, pEMCS, and pKCS, respectively. The gene encoding a putative CitS was expressed in a pCDF-derived plasmid under the control of the *trc* promoter, which was designated as pKCT.

### Transformation of NG13

Overnight pre-cultured NG13 cells were inoculated in fresh LB medium and cultured with shaking at 30 °C until they reached the early exponential phase (OD$_{660}$ = 0.4). Cells were harvested and washed with cold distilled water, followed by 10% glycerol. Cells were then resuspended in 10% glycerol and stored at −80 °C until further use. Regarding transformation, a 60-μL aliquot of cells was thawed on ice, mixed with plasmid DNA (0.1 μg), and kept on ice for 30 min. The cell suspension was subjected to electroporation using a GENE PULSER II (Bio-Rad, Hercules, CA, USA) at 25 kV cm$^{-1}$, 25 μF, and 200 Ω. After electroporation, cells were transferred into 1 mL LB medium without antibiotics, shaken at 30 °C for 1 h, and plated onto an LB agar plate containing tetracycline (3 μg mL$^{-1}$) and/or spectinomycin (100 μg mL$^{-1}$) to obtain transformants.

### $^{15}$N and $^{13}$C incorporation assays

In the $^{15}$N incorporation assay, cells were inoculated into 5 mL KDC medium in φ25 test tubes, which were stoppered with butyl-rubber, and the gas phase was replaced with a mixture of $^{15}$N$_2$:O$_2$ (82:18) (Shoko Science, Kanagawa, Japan) for the $^{15}$N$_2$ sample or air for the $^{14}$N$_2$ sample. Test tubes were statically incubated at 25 °C for 6 d. In the $^{13}$C incorporation assay, cells were inoculated in KDC medium in which glucose or citrate was replaced with $^{13}$C$_6$-glucose (>99 atom % $^{13}$C) or 1,5-$^{13}$C$_2$-citrate (98 atom % $^{13}$C), respectively. Culturing was performed in 5 mL of medium in non-hermetic φ25 test tubes at 25 °C for 6 d. To analyze glutamate in the supernatant, the culture supernatant was diluted 10-fold with methanol and transferred to a

UFLC Nexera system (Shimadzu) connected to a Triple TOF 5600 system (SCIEX, Tokyo, Japan). Samples were separated using an ACQUITY UPLC BEH HILIC column, 130 Å, 1.7 μm, 2.1 × 50 mm (Waters, Milford, MA, USA). LC conditions were as follows: mobile phase A, $H_2O$ + 0.1% formic acid; mobile phase B, acetonitrile + 0.1% formic acid; 95% B for 1 min, 95%–40% B over 5 min, 40% B for 3 min, and 95% B for 3 min, at a flow rate of 0.4 mL min$^{-1}$. Ions of 2.7 ± 0.2 min (retention time of standard glutamate) were analyzed. MS analyses were simultaneously performed using electrospray ionization in the positive mode.

### Glutamate productivity

Glutamate productivity was calculated according to (1) glucose-based productivity and (2) citrate-based productivity. (1) Glucose-based productivity is based on the consumption of glucose to produce ATP for glutamate production. As described in Fig. 1, to synthesize 1 mole of glutamate, 0.5 mole of aerial nitrogen is fixed by nitrogenase with the consumption of 8 moles of ATP and fixed ammonia is assimilated into glutamate by the GS-GOGAT system with the consumption of 1 mole of ATP. A total of 9 moles of ATP are supplied by the Embden–Meyerhof–Parnas pathway consuming 4.5 moles of glucose in *K. oxytoca*. When 100% productivity is achieved, 1.36 g L$^{-1}$ (9.3 mM) and 2.72 g L$^{-1}$ (18.5 mM) glutamate may be produced in the KDC (41.6 mM glucose) and KDDC (83.3 mM glucose) medium, respectively. (2) Citrate-based productivity is a citrate-to-glutamate conversion ratio based on the results showing that the CitS and CgCS+CitS strains produced glutamate only from citrate. One mole of glutamate is theoretically produced from 1 mole of citrate as a precursor; therefore, productivity of 100% indicates that all the consumed citrate was converted into glutamate produced.

### Statistics and reproducibility

All the other experiments were performed at least three times on independent days. Data are shown as the mean of three biological replicates with standard deviation. Data were analyzed using Microsoft Excel version 2021.

### Reporting summary

Further information on research design is available in the Nature Portfolio Reporting Summary linked to this article.

## Data availability

All data in this study are included in the article and its Supplementary information. Numerical source data for all graphs and tables can be found in Supplementary Data. Newly sequenced genes encoding KoCS and CitS have been deposited in DDBJ under accession numbers LC804971 and LC804972, respectively. Materials and all other primary data files are available upon reasonable request from the corresponding authors.

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

## Acknowledgements
We thank H. Kawasaki for advice regarding bacterial carbon metabolism for amino acid production. We thank T. Shiraishi for instructions on LC–MS analysis. This work was supported by a research grant from Kikkoman Corporation.

## Author contributions
M.N. and D.Y. are the co-corresponding authors of this study. M.N. and M.H. conceived and designed the project. S.K., M.N., and Y.I. performed the project. D.Y., M.H., S.K., and M.N. designed the experiments. D.Y. constructed plasmids, generated and cultured *K. oxytoca* mutants, and measured culture components. T.M. performed LC–MS analyses. D.Y., M.H., S.K., and M.N. wrote and edited the manuscript with input from all authors.

## Competing interests
The authors declare the following competing interests: Y.I. is an employee of Kikkoman Corporation. D.Y., Y.I., K.S., M.H., and M.N. have filed patent applications on this work. M.N. received research funding from Kikkoman Corporation. T.M. declares no competing interests.
