## [Peer Review File · Communications Biology]

Reviewers' comments:

Reviewer #1 (Remarks to the Author):

COMMSBIO-23-3641-T describes the development of a microbial biocatalyst for glutamate production from aerial nitrogen. It is well-written but lacking in excitement. My interpretation is that the main novelty lies in demonstrating biosynthesis of a nitrogen-containing product using a nitrogen-fixing organism. They describe that this has been achieved before, but with poor titer. Hence, the novelty here is that they obtain higher titer, but they haven't explained why this is the most critical parameter for a financially feasible manufacturing process - i.e. if separation is a dominant cost factor.

Anyway, what will it save? Approximately throughout, assuming global glutamate production requires 300,000 tonnes of sugar, this will consume 2m tonnes of nitrogen fertiliser just to produce sugar, without considering nitrogen supplied as substrate, all to save 15,000 tonnes of N by using a N₂-fixer? So, the big problem to solve is carbon, not nitrogen. Hence, a carbon fixing system is more likely to reduce the demand for Haber-Bosch. To give context, this Klebsiella technology would reduce Haber-Bosch demand by 0.0001%, assuming that the technology fully outcompetes all other technologies on the market. A drop in the ocean.

It is unclear if the improvement in the conditions that was used to obtain the highest titer as different units are used for the same measure, i.e. cell number per tube vs. plate, which is totally unhelpful for any meaningful comparison.

A map of the relevant metabolism would be a most suitable starting point, not at the end. The engineering design also needs to be illustrated, that is critical.

Fig. 1, some data points overlap so it is not visible. This needs to be fixed. Describe the strains in a meaningful way in the legends so as to avoid having readers remember meaningless acronyms. The O₂ content of the culture needs to be described in the legend.

What is different between Fig. 5 and Fig. 2, it should be obvious without searching for information elsewhere.

Fig. 6 needs a legend. How was it decided what parts of metabolism to retain and which to delete? I can't understand this figure. Full stoichiometry and context is needed.

Reviewer #2 (Remarks to the Author):

Glutamate is an indispensable component in all living organisms, and widely applied in different areas, while its biological production mainly depends on extra N source supply. Using N resource derived by BNF from N-fixing bacterium would be an eco-friendly way to produce glutamate. The study by Yoshidome et al. focused on Klebsiella oxytoca, and made great efforts to improve its nitrogenase activity and glutamate yield by optimizing the carbon source and incubation conditions, and genetically engineering to overexpress the genes encoding citrate synthase and Na⁺-dependent citrate symporter. Finally, the study developed an eco-friendly method for glutamate production, which I think is a very interesting work with novel idea and good outcomes. I read it with keen interests. The study is well designed, and I noticed that the revised MS addressed the main questions posed by previous reviewers. I only have some minor comments as below.

1. Line 78, the sentence is not complete. Line 80-83, it would be better to express in a way similar as in line 84 (The third) to state the research contents and aims, rather than describing the results directly.

2. I have similar concern on the unit of nitrogenase activity like " $\mu\text{mol 4-h-1 tube-1}$ ". I think it is hard to compare as the study used tube and plate with different volumes. Why not express as per OD, or per ml culture? In contrast, line 351 and 352 were expressed as g L-1 OD660-1 , which is abrupt. Unify them would be better. Further, why the optical density is measured at wavelength 660nm rather than 600nm?

3. Line 341, 349-350, supplement the methods on how the energy conversion efficiency was calculated in Methods and Materials section.

4. I have similar concern on why *Klebsiella oxytoca* was used in this study, not other diazotrophic bacteria. I saw authors addressed this question clearly in the rebuttal file. I suggest to add this information in somewhere in introduction or discussion for reader's reference.

Reviewer #1 (Remarks to the Author):

COMMSBIO-23-3641-T describes the development of a microbial biocatalyst for glutamate production from aerial nitrogen. It is well-written but lacking in excitement. My interpretation is that the main novelty lies in demonstrating biosynthesis of a nitrogen-containing product using a nitrogen-fixing organism. They describe that this has been achieved before, but with poor titer. Hence, the novelty here is that they obtain higher titer, but they haven't explained why this is the most critical parameter for a financially feasible manufacturing process - i.e. if separation is a dominant cost factor.

Anyway, what will it save? Approximately throughout, assuming global glutamate production requires 300,000 tonnes of sugar, this will consume 2m tonnes of nitrogen fertiliser just to produce sugar, without considering nitrogen supplied as substrate, all to save 15,000 tonnes of N by using a N₂-fixer? So, the big problem to solve is carbon, not nitrogen. Hence, a carbon fixing system is more likely to reduce the demand for Haber-Bosch. To give context, this Klebsiella technology would reduce Haber-Bosch demand by 0.0001%, assuming that the technology fully outcompetes all other technologies on the market. A drop in the ocean.

We would like to express our sincere thanks to this reviewer for providing us valuable comments.

The energy involved in the process of glutamate formation is currently based on a novel method called "L-glutamate crystallization fermentation"¹. This method is based on the property that the solubility of glutamate decreases under acidic conditions. Since glutamate is separated from the culture as crystals, this method saves a lot of energy used in previous separation. We expect that L-glutamate crystallization is also applicable to our method. In addition, our glutamate-producing method is conducted in a static culture under air at room temperature (25°C), not requiring energy-consuming process such as heating, agitation, and gas supply.

We also would like to thank this reviewer for give us comments on economical and economical impact of our new glutamate-producing method. We fully agree to the comments by this reviewer stating that CO₂ fixation is the most concern to be solved in the sustainable society in the very near future, which has been added in the revised manuscript (Line 45). However, we also think that reduction of the Haber-Bosch process should not be ignored as follows. The total amount of N fixed by the Haber-Bosch process is estimated to be 120 million tons per year, of which 96 million tons is used as nitrogen fertilizer and 24 million tons is used industrially including glutamate fermentation². About 4% of the nitrogen fertilizer (4 million tons of N), is used in the production of sugar crops, producing 179 million tons of sugars^{3,4}.

Glutamate of about 4 million tons is produced in a year⁵, therefore about 5 million tons of sugar (about 2.8% of total sugar production, so about 0.11 million tons of N is used as chemical fertilizer for this production) and 0.5 million tons of ammonium are needed as raw materials without considering loss of supplied sugar and ammonium during glutamate-producing process. Hence, more N as ammonium supply is required in glutamate production than N as fertilizer to produce sugars used for glutamate production. Furthermore, it is clear that glutamate production from aerial nitrogen would lead to a reduction in Haber-Bosch demand, which would correspond to 0.4% of the total Haber-Bosch process (especially 2.1% in industrial use). We think this amount can never be ignored, although the current increase in carbon demands is also very crucial.

It is unclear if the improvement in the conditions that was used to obtain the highest titer as different units are used for the same measure, i.e. cell number per tube vs. plate, which is totally unhelpful for any meaningful comparison.

All units have been standardized by culture volume instead of per plate or per tube.

A map of the relevant metabolism would be a most suitable starting point, not at the end. The engineering design also needs to be illustrated, that is critical.

Fig. 6 in the original manuscript that shows a map of the relevant metabolisms was used as Fig. 1 in the revised manuscript with detailed legend added and the explanation of the metabolisms and stoichiometry were also provided in the legend.

Fig. 1, some data points overlap so it is not visible. This needs to be fixed. Describe the strains in a meaningful way in the legends so as to avoid having readers remember meaningless acronyms. The O₂ content of the culture needs to be described in the legend.

We added note on the overlap of some data points in the legend. Also, description of acronyms of strain names were added in the legend. The O₂ content of the culture was not measured because we did not have a dissolved oxygen meter. Instead, we revised the legends to state that the culture in this study was conducted with a constant O₂ supply.

What is different between Fig. 5 and Fig. 2, it should be obvious without searching for information elsewhere.

ϕ 25 test tubes were used in Fig. 2 (original manuscript) and ϕ 90 plates were used in Fig. 5 (original manuscript) for examining the effect of air supply on glutamate production. For clarity, we added information on the culture vessel used in each culture at the top of each figure in the revised manuscript.

Fig. 6 needs a legend. How was it decided what parts of metabolism to retain and which to delete? I can't understand this figure. Full stoichiometry and context is needed.

Fig. 6 in the original manuscript was revised and treated as Fig. 1 in the revised manuscript. In the new Fig. 1, we provided the stoichiometry of the metabolism and revised a legend for readers to understand the relevant metabolisms described in this manuscript. Note that in this study, we have not disrupted any genes originally present in the NG13 strain, so only the metabolism presumably involved in glutamate production is described.

Reviewer #2 (Remarks to the Author):

Glutamate is an indispensable component in all living organisms, and widely applied in different areas, while its biological production mainly depends on extra N source supply. Using N resource derived by BNF from N-fixing bacterium would be an eco-friendly way to produce glutamate. The study by Yoshidome et al. focused on *Klebsiella oxytoca*, and made great efforts to improve its nitrogenase activity and glutamate yield by optimizing the carbon source and incubation conditions, and genetically engineering to overexpress the genes encoding citrate synthase and Na⁺-dependent citrate symporter. Finally, the study developed an eco-friendly method for glutamate production, which I think is a very interesting work with novel idea and good outcomes. I read it with keen interests. The study is well designed, and I noticed that the revised MS addressed the main questions posed by previous reviewers. I only have some minor comments as below.

We would like to thank this reviewer for the kind comments.

1. Line 78, the sentence is not complete. Line 80-83, it would be better to express in a way similar as in line 84 (The third) to state the research contents and aims, rather than describing the results directly.

We revised them in the revised manuscript (Lines 79-81, 89-93).

2. I have similar concern on the unit of nitrogenase activity like "μmol 4-h-1 tube-1". I think is hard to compare as the study used tube and plate with different volumes. Why not express as per OD, or per ml culture? In contrast, line 351 and 352 were expressed as g L⁻¹ OD660-1, which is abrupt. Unify them would be better. Further, why the optical density is measured at wavelength 660nm rather than 600nm?

Since we have not yet established high density culturing system for NG13, the cell growth during glutamate production in this study was extremely low compared with that in the

current *C. glutamicum* culture (added this to Lines 354-357 in the revised manuscript). Therefore, we thought that it would not be meaningful to compare the glutamate productivity by g L^{-1} , and used $\text{g L}^{-1} \text{OD}_{660}^{-1}$ in lines 351-352 of the original manuscript. Anyhow, we modified the all units of nitrogenase activity to $\mu \text{mol 4-h}^{-1} \text{mL}^{-1}$ in all the figures in the revised manuscript. Since we could only find papers that described glutamate production in *C. glutamicum* where the cell growth was monitored by OD_{660} , we used the OD_{660} of *K. oxytoca* only for this part. Although we know that OD_{600} is more often used as OD unit, OD_{660} is sometimes used and there is a correlation between the cell count and OD_{660} in *E. coli* and other bacteria⁶.

3. Line 341, 349-350, supplement the methods on how the energy conversion efficiency was calculated in Methods and Materials section.

We have added the detailed methods on how the glutamate productivity was calculated in the Methods section (Lines 477-492 in the revised manuscript).

4. I have similar concern on why *Klebsiella oxytoca* was used in this study, not other diazotrophic bacteria. I saw authors addressed this question clearly in the rebuttal file. I suggest to add this information in somewhere in introduction or discussion for reader's reference.

In Introduction section, we added the reason why *Klebsiella oxytoca* was used in this study like addressed in the rebuttal file (Lines 79-89).

References

1. Usuda, Y., Hara, Y. & Kojima, H. Toward sustainable amino acid production. *Adv. Biochem. Eng. Biotechnol.* **159**, 289–304 (2017).
2. Fowler, D. *et al.* The global nitrogen cycle in the Twentyfirst century. *Philos. Trans. R. Soc. B Biol. Sci.* **368**, (2013).
3. Ludemann, C. I., Gruere, A., Heffer, P. & Dobermann, A. Global data on fertilizer use by crop and by country. *Sci. Data* **9**, 1–8 (2022).
4. ISO Sugar Yearbook 2023. (2023).
5. Sanchez, S., Rodríguez-Sanoja, R., Ramos, A. & Demain, A. L. Our microbes not only produce antibiotics, they also overproduce amino acids. *J. Antibiot.* **71**, 26–36 (2018).
6. Pan, H., Zhang, Y., He, G. X., Katagori, N. & Chen, H. A comparison of conventional methods for the quantification of bacterial cells after exposure to metal oxide nanoparticles. *BMC Microbiol.* **14**, (2014).

Reviewers' comments:

Reviewer #1 (Remarks to the Author):

The purpose of the review is not to have a closed internal discussion, but rather to inspire or request changes to the manuscript. I would have hoped that the first point made regarding economics and scale would have resulted in amendments that help provide the reader with a sense of context. I don't see such a change in the manuscript. Furthermore, there is no clarification why improved titer is of interest as a key goal? At the end of the discussion the focus is instead on productivity (which makes much more sense!). I strongly suggest to change the parameter throughout (incl. abstract), from titer to productivity. Still, the choice of parameter has to be explained and argued in the context of impact - i.e. what matters to eventual realisation and impact to society.

It is welcome to see Fig. 6 move to 1. It is still somewhat unintuitive and overwhelming but it helps the reader get a feeling for the work at the outset.

Lines 85-87 is not understandable. "the carbon flux to glutamate were expected not to deteriorate its nitrogen fixation"????? This is a combined issue with both language and technical detail.

Reviewer #2 (Remarks to the Author):

The authors addressed my comments and questions clearly, and I have no further comment.

R1

The purpose of the review is not to have a closed internal discussion, but rather to inspire or request changes to the manuscript. I would have hoped that the first point made regarding economics and scale would have resulted in amendments that help provide the reader with a sense of context. I don't see such a change in the manuscript.

-We thank you for your variable suggestion to improve our manuscript. We have described that the amount of chemically fixed nitrogen by the Haber-Bosch process is consumed by the current industrial glutamate fermentation by considering both direct use in ammonia production as its nitrogen source and indirect use in nitrogen fertilizer production to obtain sugar as its carbon source in the introduction (Lines 39-58), which will help various readers to understand the contribution of this study to the economics more clearly.

The resulting sentences (Lines 39-58) are:

“To meet the demand for practical applications, a large amount of glutamate (3.3 million tons (Tg) yr⁻¹) is produced industrially using microorganisms⁵. Glutamate fermentation was initially established by Kinoshita and Udaka using the aerobically grown bacterium *Corynebacterium glutamicum*⁶. However, this process requires a large amount of ammonium and glucose as nitrogen and carbon sources, respectively, which are directly or indirectly dependent on chemically fixed nitrogen of 120 Tg yr⁻¹ via the Haber–Bosch process⁷, which is highly energy-demanding (approximately 2% of total energy) and emits large amounts of CO₂⁸. To produce 3.3 Tg yr⁻¹ glutamate, approximately 0.31 Tg yr⁻¹ of fixed nitrogen and 3.84 Tg yr⁻¹ of sugar are stoichiometrically required as raw materials. Fixed nitrogen of 4.03 Tg yr⁻¹ is applied as nitrogen fertilizer to sugar crops worldwide⁹ in order to produce around 170 Tg yr⁻¹ of sugar in the world¹⁰. Therefore, 0.095 Tg yr⁻¹ fixed nitrogen, which corresponds to 0.08% of total Haber-Bosch demands, is estimated to be indirectly consumed for glutamate fermentation via its carbon source, while 0.31 Tg yr⁻¹ fixed nitrogen as a nitrogen source for glutamate fermentation corresponds to 0.26%. In total, 0.34% of nitrogen chemically fixed by the Harber-Bosch process is used for current glutamate fermentation. Although this value appears to be low; it only reflects glutamate fermentation; therefore, a sustainable and eco-friendly bioprocess independent of the Harber-Bosch process needs to be established in the near future.”

Furthermore, there is no clarification why improved titer is of interest as a key goal? At the end of the discussion the focus is instead on productivity (which makes much more sense!). I strongly suggest to change the parameter throughout (incl. abstract), from titer to productivity. Still, the choice of parameter has to be explained and argued in the context of impact - i.e. what matters to eventual realisation and impact to society.

-Firstly, we apologize for the ambiguous definition of the word "productivity". We used "productivity" for g L^{-1} (Lines 355, 358 in the original manuscript), $\text{g L}^{-1} \text{OD}_{660}^{-1}$ (Line 359 in the original manuscript), and % (ATP consumption rate for glutamate production and citrate-to-glutamate conversion rate, in the original Method section). Considering that the original meaning of "productivity" is the production rate of the amount of materials necessary to produce the goods, we have revised the usage of "productivity" only for % (ATP consumption rate for glutamate production and citrate-to-glutamate conversion rate) and changed these descriptions to glucose-based glutamate productivity and citrate-based glutamate productivity, respectively. Also, we have revised the description of these productivities in Method section (Lines 507-521). We would like to note that the glucose-based glutamate productivity of the CgCS strain has been deleted in the revised Table 1. We have noticed that the amount of ATP production could not be simply estimated in the CgCS strain under the culture containing glucose as a sole carbon source, because glucose is used as both the energy source and glutamate skeleton in the strain. We think that the deletion of the data would not have affected the quality of this manuscript.

Corresponding portions in the revised manuscript are:

Lines 368-382:

"We calculated the productivity of glutamate based on the consumption ratio of glucose to produce ATP for glutamate production, assuming that 4.5 moles of glucose are theoretically consumed to synthesize one mole of glutamate (see Methods section for details). Based on this assumption, 1.36 g L^{-1} (9.24 mM) glutamate may be maximally produced in the KDC (containing 41.6 mM glucose) medium by the CgCS+CitS strain, which was used to evaluate glucose-based productivity. Glutamate production of 78 mg L^{-1} (0.53 mM) and 335 mg L^{-1} (2.28 mM) by the CitS and CgCS+CitS strains, respectively, under $\phi 25$ -KDC conditions corresponded to 5.8 and 24.6%, respectively, in glucose-based productivity (Table 1). Productivity of the CgCS+CitS strain was further increased to 44.5 % under $\phi 90$ -KDC condition and 41.4% under $\phi 90$ -KDDC condition. We also calculated glutamate productivity according to the conversion ratio from citrate to glutamate (citrate-based productivity) in the CitS and CgCS+CitS strains (Table 1). Citrate-based productivity increased from 10.7% under $\phi 25$ -KDC condition by the CitS strain to 81.7% under $\phi 90$ -KDDC condition by the CgCS+CitS strain."

Lines 507-521:

"Glutamate productivity"

Glutamate productivity was calculated according to 1) glucose-based productivity and 2) citrate-based productivity. 1) Glucose-based productivity is based on the consumption of glucose to produce ATP for glutamate production. As described in Fig 1, to synthesize one mole of glutamate, 0.5 mole of aerial nitrogen is fixed by nitrogenase with the consumption of 8 moles of ATP and fixed ammonia is

assimilated into glutamate by the GS-GOGAT system with the consumption of 1 mole of ATP. A total of 9 moles of ATP are supplied by the Embden–Meyerhof–Parnas pathway consuming 4.5 moles of glucose in *K. oxytoca*. When 100% productivity is achieved, 1.36 g L⁻¹ (9.3 mM) and 2.72 g L⁻¹ (18.5 mM) glutamate may be produced in the KDC (41.6 mM glucose) and KDCC (83.3 mM glucose) medium, respectively. 2) Citrate-based productivity is a citrate-to-glutamate conversion ratio based on the results showing that the CitS and CgCS+CitS strains produced glutamate only from citrate. One mole of glutamate is theoretically produced from one mole of citrate as a precursor; therefore, productivity of 100% indicates that all the consumed citrate was converted into glutamate produced.”

Next, we carefully re-discussed the reasonable parameter for describing glutamate production in terms of the impact on the society and economy. In the actual fermentation, both the titer (g L⁻¹) and the rate of the production (g L⁻¹ h⁻¹) are important factor for the separation process and cost for fermentation due to heating or shaking, respectively. Therefore, we remained the titer in the results and have added the rate of glutamate production to all results in the text and Table 1. In the original manuscript, we described the glutamate productivity per cell (g L⁻¹ OD₆₆₀⁻¹) in our study surpassed that of *C. glutamicum*, which would give the impression as if our new method could immediately replace the production system using *C. glutamicum*. However, the rate of the glutamate production in our methods was more than 300 times lower than that of *C. glutamicum*. Therefore, it is obvious that we still have a long way to implement our method to the society. Reducing the required culture time and increase in the cell density in the culture are concerned issues to be improved in the future . We have described this more clearly in the revised manuscript (Lines 383-394).

The revised portions (Lines 383-394) are:

“Approximately 20 g L⁻¹ glutamate was produced by *C. glutamicum* using ammonium, while our method generated nearly 1 g L⁻¹, which was still low. This difference was attributed to the lower cell number corresponding to OD₆₆₀ of 0.29 in our 1 g L⁻¹ culture with a ϕ90 plate than that of *C. glutamicum* (around OD₆₆₀ = 40)⁴¹. When the glutamate production per cell was compared between the *C. glutamicum* system and ours, ours (3.8 g L⁻¹ OD₆₆₀⁻¹) surpassed that of *C. glutamicum* (0.58 g L⁻¹ OD₆₆₀⁻¹). In terms of the production rate, the production of a nitrogen-containing compound at approximately 3 mg L⁻¹ h⁻¹ was achieved in our culture system. Although our glutamate-producing rate was markedly lower than that of *C. glutamicum* (952 mg L⁻¹ h⁻¹), our fermentation process does not require an energy-consuming process, such as shaking or air supply. Future studies are required to investigate approaches to reduce the required culture time and increase cell number in the culture.”

Production rates are provided in the revised manuscript.

In Lines 179, 187, 209, 211, 214, 232, 235, 292, and 293

It is welcome to see Fig. 6 move to 1. It is still somewhat unintuitive and overwhelming but it helps the reader get a feeling for the work at the outset.

-We appreciate your suggestion to move the outline of this study to Fig. 1. To further improve the readability of Fig. 1 for various readers, we have categorized the metabolism pathway into four sections (glucose metabolism, TCA cycle, nitrogen fixation, and nitrogen assimilation) and highlighted them in the revised version of Fig. 1. Although NAD(P)H/NAD(P)⁺ ratio is very important to optimize bioproduction, we did not focus on the ratio in the present study; therefore we have also deleted the description of NAD(P)H/NAD(P)⁺. We have also revised the legend for more readability.

Lines 85-87 is not understandable. "the carbon flux to glutamate were expected not to deteriorate its nitrogen fixation"?????? This is a combined issue with both language and technical detail.

- We revised them in the revised manuscript (Lines 91-99).

These (Lines 91-99) are:

"1) ATP and reducing power for nitrogen fixation are provided by glycolysis in *K. oxytoca*²⁴, which is different from other diazotrophs such as *Azotobacter* or *Azospirillum* species that depend on the tricarboxylic acid (TCA) cycle to obtain energy components for nitrogen fixation^{25,26}. This is a key point for glutamate production from aerial nitrogen because glutamate is synthesized from 2-OG, a compound composing the TCA cycle, and, thus, the enhancement of carbon flux from 2-OG to glutamate generally competes with the TCA cycle to obtain energy for nitrogen fixation. *K. oxytoca*, which fixes nitrogen independent of the TCA cycle, is not expected to cause metabolic conflict and is compatible with glutamate production."

R2

The authors addressed my comments and questions clearly, and I have no further comment.

-We would like to thank you for your agreement.